# Database of Potential Promoter Sequences in the *Capsicum annuum* Genome

**DOI:** 10.3390/biology11081117

**Published:** 2022-07-26

**Authors:** Valentina Rudenko, Eugene Korotkov

**Affiliations:** Institute of Bioengineering, Research Center of Biotechnology of the Russian Academy of Sciences, Moscow 119071, Russia; bioinf@yandex.ru

**Keywords:** *Capsicum annuum*, pepper genome, potential promoter sequences, promoter classification, promoter prediction, plant promoter database, MAHDS

## Abstract

**Simple Summary:**

In this study, we searched for potential promoter sequences (PPS) in the pepper genome. We used a new mathematical method for the multiple alignment of highly divergent sequences. Hence, 20 statistically significant classes of sequences in the range from −499 to +100 nucleotides near the annotated genes were calculated. A profile was constructed for each class, which was then used as a position–weight matrix to build a two-dimensional alignment. We found 825,136 potential promoter sequences with a false positive rate of 0.13% in the pepper genome. They were subsequently merged into a database. Potential promoter sequences were analyzed by TSSFinder software, which detected transcription start sites in more than a half of our data. The results show that the pepper genome contains many PPSs. We assume that most of them could be associated with various transposons, dispersed repeats, or viruses.

**Abstract:**

In this study, we used a mathematical method for the multiple alignment of highly divergent sequences (MAHDS) to create a database of potential promoter sequences (PPSs) in the *Capsicum annuum* genome. To search for PPSs, 20 statistically significant classes of sequences located in the range from −499 to +100 nucleotides near the annotated genes were calculated. For each class, a position–weight matrix (PWM) was computed and then used to identify PPSs in the *C. annuum* genome. In total, 825,136 PPSs were detected, with a false positive rate of 0.13%. The PPSs obtained with the MAHDS method were tested using TSSFinder, which detects transcription start sites. The databank of the found PPSs provides their coordinates in chromosomes, the alignment of each PPS with the PWM, and the level of statistical significance as a normal distribution argument, and can be used in genetic engineering and biotechnology.

## 1. Introduction

Advances in genome sequencing have led to the emergence of a huge amount of data and, as a result, to the creation of biological databases [1]. Genbank was one of the first databases to appear that contains all the available DNA and protein sequences. The data contained in biological databases can be conditionally divided into two types: verified (annotated) and raw. Accessible format of this data allows user to find necessary biological information, which greatly saves time and resources. Among the biological databases used, primary and secondary databases can be distinguished. Information about DNA and protein sequences, as well as protein structures information, is contained in primary databases. These databases include GenBank and DDBJ, Swiss-Prot and PIR, as well as Protein Data for protein structure. Secondary databases contain information processed in one way or another from primary databases. Particularly relevant are secondary databases with biological information processed using new algorithms. Such analysis often makes it possible to obtain new information, which can be very important, since similar experimental data is not available.

Bell pepper (*Capsicum annuum*) is an important agricultural crop. Peppers belong to the Capsicum genus, which contains over 20 species and belongs to the Solanaceae family comprising many other agricultural species, including tomato and potato. Peppers are known for their beneficial health effects, because they contain vitamins, minerals, flavonoids, carotenoids, and capsaicin, responsible for a burning sensation of peppers, which is used to treat neuralgia [2]. Peppers have a diploid genome of over 3 GB in size, which exceeds that of the human genome and encodes about 34,900 genes. Furthermore, it has been estimated that more than 81% of the pepper genome consists of mobile elements such as transposons and retrotransposons [3]. However, for a complete annotation of the *C. annuum* genome, it is important to also identify all promoter regions, which can point on the genes not detected by RNA sequencing, potential transcription start sites (TSSs), and transposons. 

Gene promoter regions, which are necessary to initiate the process of DNA transcription to RNA, are located in the region from −499 to +100 bp (where +1 is the TSS position) and contain binding sites for RNA polymerase and transcriptional regulators [4]. It is known that eukaryotic promoters contain highly conserved motifs, including the initiator (Inr), TATA-box, and downstream promoter element (DPE). Important motifs downstream of the TSS, such as motif ten element (MTE) and Bridge, a bipartite core promoter element, have also been described [5].

Computer methods to search for promoters are being developed intensively in the last 20 years. Although there are many algorithms to identify DNA coding regions [6], the detection of promoter sequences increases the reliability of these algorithms and enables determination of the TSS position. The information about the location of promoter regions has important practical applications in genetic engineering, because gene expression is controlled through the regulation of promoter activity [7,8]. 

Advances in sequencing technologies [9] make it possible to accurately determine the position of TSSs, which could be found experimentally using OligoCap, CAGE, and deepCAGE [10,11]. However, these methods are difficult to implement and require highly qualified staff and modern DNA sequencers. Therefore, in most cases, potential promoters are detected by bioinformatics rather than experimental methods [12]. The in silico identified promoter sequences are accumulated in databases, such as the Eukaryotic Promoter Database (EPD) [13,14]. However, although all promoters deposited in the EPD have been experimentally confirmed, this has been performed in model organisms, whereas promoters in the other species may be quite different.

The mathematical approach of pattern recognition, implemented as neural networks, shows good results in finding promoter sequences, if the network is trained on the existing set of promoters taken from a database [15]. Other methods, such as context-free grammars [16], are also successfully applied in cases when the structure and localization of individual promoter motifs are known. However, the challenge of identifying promoter regions lies in their extreme diversity. Even prokaryotic promoters are classified depending on respective sigma factors, and the acceptable quality of promoter prediction can be achieved by customizing search methods for a specific class [17,18]. Therefore, a problem that cannot be solved by the currently developed mathematical methods is the generation of a large number of false positives. Thus, even the best promoter search programs produce one false positive result per 20,000 bp [19,20,21,22,23,24,25], which, if extrapolated to a whole genome of about 3 × 10^9^ bp, yields more than 300,000 false positives. Given that such a genome contains over 10^4^ genes, it means that the number of false positives exceeds that of gene promoters by several times. Therefore, to reliably identify promoter sequences in eukaryotic genomes, it is necessary to develop novel mathematical algorithms that generate a low number of false positive results.

Previously, we have developed a method for the multiple alignment of highly diverged sequences (MAHDS) based on a genetic algorithm and two-dimensional dynamic programming [26]. The main advantage of the method is that it finds statistically significant multiple alignments without pairwise alignment of the analyzed sequences. During the execution of the method, a random pattern of multiple alignments is optimized using a genetic algorithm. Thus, the position–weight matrix (PWM), which is considered as a pattern, is optimized with the aim to find a PWM showing the best two-dimensional alignment with the analyzed sequences. The use of PWMs makes it possible to calculate multiple alignments for highly divergent sequences, which have accumulated up to 4.4 substitutions per nucleotide. When the MAHDS method was applied to construct multiple alignments of promoter sequences of *Arabidopsis thaliana* and *Oryza sativa* genomes, the number of false positives was reduced to 10^−8^ per nucleotide, i.e., to no more than 100 false positives for a genome of 3 × 10^9^ bp [26,27].

Although the *C. annuum* genome has been annotated and the positions of its genes determined [3], unlike the *O. sativa* genes, there are no experimentally confirmed data on *C. annuum* gene promoters. In this work, we searched for potential promoter sequences (PPSs) in the *C. annuum* genome and created a database of the identified PPSs. This was done via the generation of promoter classes through the analysis of regions around TSSs and then applying MAHDS to find PPSs based on class matrices, which resulted in the detection of 825,136 PPSs in the *C. annuum* genome. The data are consistent with those obtained for the rice genome [27]. The PPSs identified in the *C. annuum* genome were collected in a database available at: http://victoria.biengi.ac.ru/cgi-bin/dbPPS/index.cgi (accessed on 31 March 2022). 

## 2. Materials and Methods

### 2.1. Selection and Filtering of Promoter Sequences

The genome of *C. annuum* has 12 chromosomes; their sequences and gene location information were extracted from EnsemblPlants databank (http://plants.ensembl.org/Capsicum_annuum/Info/Index (accessed on 31 March 2022)). Since there are no confirmed experimental data on *C. annuum* promoters, gene regions from −499 to +100 bp (where +1 is the TSS position) of both forward (+) and reverse (−) DNA strands were considered as promoters. According to EnsemblPlants, the total number of promoter regions was 31,600.

It appeared that some of the extracted sequences had low complexity and were enriched in repeats of one nucleotide type, in most cases A, whereas the other had poor sequencing quality, containing many ‘N’ symbols. As such sequences can significantly interfere with promoter classification, we excluded them from consideration. The promoter sequences were selected for classification according to the following criteria: (1) the content of each nucleotide was ≤60%, (2) the number of ‘N’ characters was ≤ 1 per promoter, (3) no fragments *(u_i_)_m_* consisting of one nucleotide type *u_i_* (A, T, C, or G) of length > *m*, and (4) GC composition similar to that of *A. thaliana* promoters.

Value *m_0_* was determined for each nucleotide individually. We focused on the distribution of fragments *(u_i_)_m_* among the *A. thaliana* promoters, which were confirmed experimentally. *A. thaliana* rather than *O. sativa* was chosen because our previous data indicate that its promoters are closer in nucleotide frequencies to those of *C. annuum,* probably because both *C. annuum* and *A. thaliana* are dicots. All 22,703 *A. thaliana* promoters were extracted from the EPD (https://epd.epfl.ch/arabidopsis/arabidopsis_database.php?db=arabidopsis (accessed on 31 March 2022)). In each of them, length *m* was determined for fragments *(u_i_)_m_* for all nucleotide types. It turned out to be the case that the probability of fragments (A)_m_ for *m* > 13 was less than 4.24%, (T)_m_ for *m* > 12 was less than 3.65%, and (C)_m_ and (G)_m_ for *m* > 5 were less than 1.11% and 0.76%, respectively. Therefore, we chose level *m_0_* equal to {13, 12, 5, 5} for bases {A, T, C, G} respectively, and accordingly, removed the sequences having at least one fragment *(u_i_)_m_* with *m* > *m_0_* from the 31,600 promoter regions of the *C. annuum* genome. Then, we performed analysis of the GC composition of *A. thaliana* promoters, which revealed that the frequencies of G and C symbols were in the range from 0.29 to 0.37. Therefore, to form the training set, we also removed the promoter regions with a GC frequency <0.29 or >0.37. 

Finally, after applying all filters, a set of 16,285 *C. annuum* promoter sequences remained for further analysis. Among them, 8115 and 8170 were located on the forward and reverse strands, respectively (Appendix A). This set was denoted as *Set_1_*, and the number of included sequences as *N_s_*.

### 2.2. Construction of Promoter Sequence Classes

The promoter sequences from *Set_1_* were classified, and a PWM was created for each class. Promoter classes were formed iteratively as described earlier [25,26]. To create classes, we randomly selected 1000 sequences from *Set_1_* and searched for their multiple alignment using the MAHDS algorithm. For that, the online tool http://victoria.biengi.ac.ru/mahds/auth (accessed on 31 March 2022) was used. As a result, we obtained multiple alignment *V(i,j)*, where *i* was the sequence number from 1 to 1000, and *j* was the column number from 1 to *L_1_*. The alignment length was equal to *L_1_*, which may not be equal to the length of the aligned sequences because the multiple alignment contained insertions or deletions (indels). Then, we built *PWM_1_*, which contained *L_1_* columns and 16 rows. The number of rows is explained by the fact that the MAHDS method constructs multiple alignments considering the correlation of adjacent DNA bases, which enables the alignment of highly divergent sequences with up to 4.4 substitutions per nucleotide [26,27]. The correlation of adjacent bases was also considered in the search for PPSs in the *C. annuum* genome (Section 2.3). 

To calculate *PWM_1_*, we first filled in frequency matrix *M(16,L_1_)*. For this, we calculated *s = v(k,j − 1) + 4(v(k,j) − 1)* for each column of multiple alignment *V* and then, added 1 to each element of matrix *M(s,j)*, which was done for all *k* from 1 to 1000 and for all *j* from 1 to *L_1_* − 1. After this, *PWM_1_* was calculated as:(1)$$PWM_{1}^{'}(i,j)=\frac{M(i,j)-N(j)p(i)}{\sqrt{N(j)p(i)(1-p(i))}}$$
where *N(j)* = $\sum_{i=1}^{16}M(i,j)$, *p(i) = Y(i)/K*, $K=\sum_{i=1}^{16}\sum_{j=1}^{L_{1}-1}M(i,j)$, $Y(i)=\sum_{j=1}^{L_{1}-1}M(i,j)$, *i* = 1, 2, .. , 16, and *j* = 1, 2, ..., *L_1_*. After that, we transformed matrix $PWM_{1}^{'}$ as described earlier [28] to obtain certain values for *R^2^* and *K_d_*, which were calculated using Formulas (2) and (3):(2)$$R^{2}=\sum_{i=1}^{16}\sum_{j=2}^{L_{1}}PWM_{1}^{'}{\left(i,j\right)}^{2}$$
(3)$$K_{d}=\sum_{i=1}^{16}\sum_{k=2}^{L_{1}}PWM_{1}^{'}(i,k)p_{1}(i)p_{2}(k)$$


Thus, matrix $PWM_{1}^{'}$ was transformed so that *K_d_* = 0 and *R^2^* = 75 *L_1_*; as a result, we obtained matrix *PWM_1_* of the first class.

Then, we searched for such sequences in *Set_1_* that had statistically significant global alignment with matrix *PWM_i_*. For this, we obtained global alignment of *PWM_i_* with each sequence from *Set_1_* and calculated similarity function *F_max_(i)* (where *i* is the sequence number in *Set_1_*, ranging from 1 to *N_s_*) at point (600, *L_1_ − 1*). Next, we randomly shuffled sequence *Set_1_(i)* and again calculated *F_r_ = F_max_(i)*; and a set of *F_r_* values with a volume of 1000 was obtained. For this set, we calculated mean value $\bar{F_{r}}$ and variance *Dis(F_r_)* and defined $Z(i)=(F_{\max}(i)-\bar{F_{r}})/{\left(Dis(F_{r})\right)}^{0.5}$ for sequence *Set_1_(i)*. In the first class, we selected only those sequences from *Set_1_* for which *Z(i) >* 5.0 and excluded them from *Set_1_*; the process of creating classes was repeated while the class volume was > 100. The program for selecting sequences from *Set_1_* with *Z*(*i*) > 5.0 is provided in the Appendix A. The process from the beginning of Section 2.2 was then repeated as we once again randomly selected 1000 sequences from *Set_1_* and searched for their multiple alignment using the http://victoria.biengi.ac.ru/mahds/auth (accessed on 31 March 2022) online tool. The procedure was terminated as soon as *PWM_i_* class volume became <100.

To select a threshold for the class volume, we classified random sequences obtained by the random mixing of all sequences from *Set_1_* and applied the classification procedure to the new set. It turned out that in this case, the class volume was approximately 16 sequences with a standard deviation of 12, indicating that the contribution of random factors in the process of classifying promoter sequences was negligible. 

### 2.3. Search for PPSs in the C. annuum Genome

After creating classes of promoter sequences, we searched for PPSs in the *C. annuum* genome. To do this, at position *k* on each chromosome (*Chr(j), j* = 1, …, 12) we allocated a window from nucleotide *k* to nucleotide *k* + *L_1_* + 49, denoted as *S(k);* the length of the window, *L_1_* + 50, was larger than that of *PWM_i_*, which enabled calculation of the alignment between *S(k)* and *PWM_i_* considering indels. Then, we built a local alignment of sequence *S(k)* with the matrix of each class *PWM_i_*, where *i* was the class number. This alignment compared sequences *S(k)* and *S_1_(l)*; the latter contained numbers from 1 to *L_1_*, which corresponded to column numbers in *PWM_i_*. The local alignment was built as described in [26]. The search for local alignment was carried out considering the correlation of neighboring DNA bases, which means that we took into account the weights of matching nucleotide pairs from *PWM_i_*.

After constructing the local alignment, the maximum value of the similarity function, denoted as *E_max_(k)*, was calculated. We determined the start and end of local alignment both in the sequence of chromosome *Chr(j)* and sequence *S_1_(l)*, denoted as (*i_0_,i_max_*) and (*l_0_, l_max_*), respectively, and searched for their local maximums *E_max_(k)* chosen because they did not intersect with each other; after that, the statistical significance of each local maximum for *PWM_i_* was evaluated. We randomly shuffled sequence *Chr(j)*, calculated *E_max_(k)* values as described above, and determined mean $\bar{E_{r}}$ and variance *Dis(E_r_)* for the series of *E_max_(k)* values to determine statistical significance $Z=(E_{\max}-\bar{E_{r}})/{\left(Dis(E_{r})\right)}^{0.5}$. As a result, we found the local maximum with Z ≥ 6.0 for each *PWM_i_* and chromosome *Chr(j)*. Next, we determined the intersection of the local maximum coordinates from different *PWM_i_*. For two local maximums with coordinates $\left(i_{0}^{1}{,}_{,}i_{\max}^{1}\right)$ and $\left(i_{0}^{2}{,}_{,}i_{\max}^{2}\right)$ and the intersection length in chromosome *Chr(j)* > 10% of the minimum ($i_{\max}^{1}-i_{0}^{1}$, $i_{\max}^{2}-i_{0}^{2}$), we chose the local maximum with the largest *Z* value. At the same time, the *PWM_i_* of the best alignment was remembered.

Thus, we generated a set of local maximums corresponding to PPSs in each chromosome. For each PPS, we obtained its coordinates (*i_0_ ,i_max_*) in the chromosome, the alignment between sequences *S(k)* and *S_1_(l)*, the *Z* value, and the corresponding *PWM_i_* and deposited these data in a database (http://victoria.biengi.ac.ru/cgi-bin/dbPPS/index.cgi).

## 3. Results

### 3.1. Promoter Sequence Classes in the C. annuum Genome

The exact number of promoters in each class is shown in Table 1. The PWMs of the classes are presented in Appendix A. The first and most representative class included about a third of all promoters. The results indicated that *C. annuum* had higher PPS diversity than *O. sativa*. Moreover, there are more PPS classes with >100 elements in *C. annuum* than in *O. sativa*. At the same time, the first five most representative classes included about 60% of all promoters both in *C. annuum* and *O. sativa* genome.

Regions from *C. annuum* genome from −499 to +100 bp, where +1 is the TSS position, were considered as promoters. Promoter positions were renumbered from 1 to 600 nt. Figure 1 shows a *PWM’* fragment (from 491 to 520 nt) for class 1. The data indicated that the weights for some nucleotide pairs significantly deviated from zero. The area comprising several nucleotides immediately before the TSS was highly conserved and did not correspond to the random distribution of dinucleotides. ATG (the start codon) was observed in position 505. Interestingly, the region downstream of the TSS has a lower frequency of occurrence of nucleotide pairs AT and TA, whereas the frequency of G-containing nucleotide pairs, i.e., TG, GA, and GG, is high.

To further investigate the conservation in promoter regions of the *C. annuum* genome, we examined base pair frequencies in the *PWM’*. For this, we first calculated:(4)$$Xi^{2}(j)=\overset{16}{\underset{i=1}{\sum }}PWM^{\prime }{\left(i,j\right)}^{2}$$

Then, $X\left(j\right)=\sqrt{2\times Xi^{2}(j)}-\sqrt{2n-1}$ (where *n* is the number of degrees of freedom, which is equal to 9, and *X(j)* is a normally distributed random variable), and plotted *X(j)* versus *j* for promoter classes 1 and 2 (Figure 2). For both classes, there was a strong peak in the TSS region: class 1 had a maximum at position 505 (*X*(505) = 180.6) and class 2 at position 503 (*X*(503) = 178.7). Analysis of *X(j)* for the PSS classes of the *C. annuum* genome revealed the following pattern: all 20 classes had peaks in the region of 495–510 bp, and the modulus *X* values for many positions were > 4, indicating that this region contained binding sites for transcription factors and other protein complexes responsible for transcriptional regulation. Figure 2 also shows the dependence of *X* on *j* for random set of PWM’ obtained by shuffling promoter sequences of class 1 and 2 as described in Section 2.2. As expected, the absolute *X(j)* values for random set of PWM’ almost never exceeded 3, whereas most *X(j)* values for the created classes 1 and 2 were over 3.

### 3.2. Clustering of Promoter Classes

A PWM for each class was created from frequency matrix *M(16,L_1_)*, which means that for each PWM constructed in Section 2.2., there was a corresponding matrix *M(16,L_1_).* The resulting *M(16,L_1_)* matrices were used for the clustering of promoter sequence classes into a smaller number of groups with the aim to speed up the search for PPSs in unannotated genomes. Since classes were designated by *M(16,L_1_)*, we expected to observe a similar distribution of dinucleotides in *M(16,L_1_)* of similar classes. At that, the global alignment weight $F^{nq}(L_{1}^{n},L_{1}^{q})$ (where *n* and *q* are numbers of matrices *M(16,L_1_)* of two classes, respectively, could serve as a quantitative measure of such similarity. The weight was defined by the following equations:(5)$$\begin{array}{l} F^{nq}\left(i,0\right)=-di,\;\\ F^{nq}\left(0,j\right)=-dj \end{array}$$
(6)$$F^{nq}\left(i,j\right)=max\left\{\begin{array}{c} F^{nq}\left(i,j-1\right)-d,\\ F^{nq}\left(i-1,j\right)-d,\\ F^{nq}\left(i,j\right)+w^{nq}(i,j) \end{array}\right\}$$
where *i* = 1, ..., $L_{1}^{n}$ and *j* = 1, ..., $L_{1}^{q}$ are column numbers in matrices *M* for promoter classes *n* and *q*, respectively; *d* is the penalty value for indels, and *w^nq^(i,j)* is the weight of the match in the alignment of the *i*^th^ column of matrix *M* for class *n* and the *j*^th^ column of matrix *M* for class *q* calculated as:(7)$$w^{nq}(i,j)=\frac{1}{4}\times \overset{16}{\underset{k=1}{\sum }}\frac{S-|m^{n}\left(i,k\right)-m^{q}\left(j,k\right)|}{\sigma }$$
where *m^n^* and *m^q^* are the elements of matrix *M* for classes *n* and *q*, respectively; *S* and *σ* are mathematical expectation and standard deviation, respectively, of a random variable representing the modulus of the difference between the elements of matrices *R^n^* and *R^q^* obtained from *M^n^* and *M^q^* by random mixing of their columns. 

Penalty *d* for indels was chosen so that in aligning any pair of available matrices *M*, the number of indels did not exceed 10. This requirement was satisfied at *d* = 20. To perform clustering, a transition from similarity functions *F^nq^* to the distance between classes was performed according to the formula:(8)$$D^{nq}={\left(\frac{\min\left(F^{nn},F^{qq}\right)-F^{nq}}{\min\left(F^{nn},F^{qq}\right)}\right)}^{2}$$

The clustering dendrogram constructed using the Complete Linkage algorithm is shown in Figure 3. To determine the level of significance for combining individual classes into groups, a similar clustering procedure was performed for random matrices *M* obtained by random mixing of rows in class matrices. All elements of the distance matrix for random matrices *M* (except for the diagonal ones) exceeded the value of 0.935. We chose an association level of 0.8 to form groups of promoter classes and obtained 10 groups (blue rectangles in Figure 3). It should be noted that five promoter classes did not cluster with any other classes.

### 3.3. Search for Potential Promoter Sequences in the C. annuum Genome

We searched for PPSs in *C. annuum* chromosomes by dynamic programming using PWMs generated for each promoter class as described in Section 2.3. The degree of similarity between the PWM and DNA sequence was defined by the *Z* value. To select threshold level *Z_0_*, we used random sequences generated by shuffling nucleotides in each chromosome. The number of PPSs in the *C. annuum* genome at different *Z* is shown in Table 2. Z Histogram for PPS is presented in Figure 4. At *Z_0_* = 6.0, 825,136 PPSs were detected; whereas for random sequences, 1068 PPSs were found at Z ≥ *Z_0_*. This result indicated that the rate of false positives was slightly more than 0.1%, i.e., we found one wrong PPS per 3 × 10^6^ DNA bases. We also examined chromosomal sequences after base substitution with complementary (without 180-degree sequence flip) and 180-degree inverted chromosomal sequences (without complementary base recoding). Let’s call these sequences forward and reverse strands. In this case, we found 92,617 sequences that have Z ≥ *Z*_0_. Consideration of this result is carried out in Section 4.

We also previously generated special sequences, where chromosome sequences were mixed in pairs or triplets. To do this, a window of 600 bases was allocated and the sequence was randomly shuffled in pairs or triplets of bases. After that, the window was moved by 300 bases and the procedure was repeated. Thus, we generated two sequences of chromosomes, one locally mixed with base pairs and the other with base triplets. In both cases, the number of similarities found for which Z ≥ *Z*_0_ did not exceed 2540. This is, of course, more than for randomly mixed sequences, but still significantly less than for 180-degree inverted sequences.

We created 100 random classes from random sequences obtained by random mixing of all sequences from *Set_1_* (see Section 2.2, last paragraph). We used these classes to search for sequences for which Z ≥ *Z*_0_. For any of the 100 classes, the number of sequences with Z ≥ *Z*_0_ was less than 100.

In order to control the accuracy of identifying promoter sequences using our method, we next determined how many of the PPSs detected at *Z* ≥ *Z_0_* coincided with the regions from −499 to +100 nt in the *C. annuum* genome. The results for each chromosome are shown in Table 3. The criterion of coincidence was the overlap between the detected PSSs and the regions near the annotated genes by at least 50%. Thus, for the first chromosome, there were 431 and 387 matches between the annotated promoter regions and PPSs on the forward (++) and reverse (−−) strands, respectively, whereas in 53 and 38 cases the promoter sequence on the forward strand coincided with that of the PPS on the reverse strand (+−) and vice versa (−+), respectively. It is expected that the numbers of (++) and (−−) cases significantly exceed those of (+−) and (−+) cases; the latter, we believe, may indicate bidirectional promoters [29].

### 3.4. Identification of PPSs in the C. annuum Genome Using TSSFinder

Many different methods and software products have been implemented for the identification of promoter sequences [17,19,30,31,32,33,34,35]. Some of them are focused on bacterial promoters and are not suitable for analysis of the *C. annuum* genome, whereas the other can identify promoter sequences with at least one false positive per 2 × 10^4^ nucleotides (verified in [26]), which means over 10^5^ false positives for the *C. annuum* genome of ~3 × 10^9^ bp. Such a high false positive rate greatly complicates quantitative comparison of PPSs found in this study with promoter sequences identified by the other methods.

A new algorithm and software package for TSS search, TSSFinder, has recently been developed based on the linear chain conditional random fields (LCCRFs) method [24]. Compared with the other algorithms, TSSFinder shows high accuracy in detecting TSSs of coding genes, whereas for the genes with a dispersed TSS signal, it determines the TSS closest to the start codon. The algorithm is flexible regarding the presence or absence of various promoter elements such as Inr, polypyrimidine initiator (TCT), TATA-box, and DPE, TFIIB recognition element and the distance between them. The TSS search program, which is available as a web server (http://sucest-fun.org/wsapp/tssfinder/ (accessed on 31 March 2022)), allows you to analyze sequences recorded in the FASTA format. There are six model organisms available as training sets for the neural network, which is also possible to train for a specific gene with known promoter sequences.

We used TSSFinder to search for TSSs among the identified PPSs. For this, we randomly selected 1200 sequences (100 from each chromosome) and tested *A. thaliana, O. sativa*, and *Homo sapiens* as model organisms. The results presented in Table 4 indicate that the *A. thaliana* genome model allows identification of the largest number of TSSs in the PPSs of the *C. annuum* genome: TSSFinder detected TSSs in more than 50% of the PPSs found in this study.

### 3.5. Search for Intersections of PPSs and Short Interspersed Nuclear Elements (SINEs) in the C. annuum Genome

We also examined the match between the found PPSs and SINE repeats in the *C. annuum* genome. Our results indicated that among 50,077 SINE repeats contained in the respective database (http://victoria.biengi.ac.ru/sine_pepper/index/ (accessed on 31 March 2022)), 9050 intersected with PPSs, and the length of the intersection was greater than 50% of that of the SINE.

### 3.6. PPS Database

The PPS database can be accessed via http://victoria.biengi.ac.ru/cgi-bin/dbPPS/index.cgi (accessed on 31 March 2022). The database uses MySQL/MariaDB for debian-linux ver 15.1. All PPSs found in the *C. annuum* genome were collected in the database, in which user can select one of 12 *C. annuum* chromosomes and search for the identified PPSs according to significance level or coordinates in the sequence. 

The database also includes PPSs found in the *O. sativa* genome [27]. It is possible to select a DNA strand for which all PPSs for each chromosome will be shown. The significance level is shown as *F_max_* and *Z* (Section 2.2) for each PPS, as well as the local alignment of the found PPS with the corresponding class matrix. For each PPS, *PWM* (Section 2.2) for which local alignment was obtained is listed. In the database, the classes for *O. sativa* are numbered from 1 to 5. A total of 20 classes were created for *C. annuum* PPS, numbering respectively 6 to 25. Each PPS in the database has its own ID. User can search the database simultaneously for the following parameters: 1. genome; 2. PPS ID; 3. PPS coordinates in the chromosome; 4. significance level of F_max_ and *Z*; 5.+ or − DNA strand; 6. class matrix. We want to expand the PPS database for other species in the future.

## 4. Discussion

The identification of potential promoter sequences is an important task for bioinformatics [36]. This is due to the fact that PPS may indicate coordinates of TSS and genes, and they can be used in genetic engineering. PPSs can also be used to analyze gene expression data and to build and understand genetic regulatory networks. In this work, the first database for potential promoter sequences obtained by computer methods was created. Previously, it was not possible to identify PPS by methods of comparing nucleotide sequences, since the promoter sequences are too different. According to our estimates, they accumulated an average of 3.6–3.7 mutations per nucleotide relative to one another [26].

The created classes of promoter sequences can reflect the participation of genes in genetic regulatory networks. It can be assumed that the more similar the classes of matrices, as shown in Figure 3, the closer to each other the genes are in the genetic regulatory network near which these promoters are located [37]. In this sense, Figure 3 is an image of the promoter-level genetic network that exists in the *C. annuum* cell. Basically, by setting a higher similarity level *Z*(*i*) (6.0 or higher) in Section 2.2, one can obtain any level of detail in the genetic network pattern that exists in the *C. annuum* genome. This opens up a new possibility for the reconstruction of genetic networks at the promoter level without using any data on the genetic activity of the genes. Here, attention is drawn to the fact that such an image can be obtained only if promoter sequences of the studied genome are known and a completely sequenced genome is used.

We estimated the number of PPS that may be active in the *C. annuum* genome. For this purpose, the presence of sequences near PPS in six transcriptomes from the *C. annuum* genome was studied. Transcriptomes were taken from https://www.ncbi.nlm.nih.gov/sra?linkname=bioproject_sra_all&from_uid=790487 (accessed on 31 March 2022). Transcriptomes were obtained from the PRJNA790487 project using Illumina HiSeq 3000. The site contains six sequence read archives (SRAs), labeled t01–t06. Let us designate the +1 nucleotide position in the found PPS in the chromosome as *TSS_pps_* coordinate; in fact, this position is a potential TSS. If the PPS performs transcription, then this position must be followed by a transcribed sequence. To verify this fact, we performed a similarity search between the 100 bp sequence located immediately after *TSS_pps_* and the sequences contained in the t01–t06 transcriptomes. The choice of 100 nucleotides was due to the fact that transcriptomes may lack RNA fragments located immediately after *TSS_pps_*. We used Megablast at NCBI website to search for similarities, since in our case we need to find 100% similarity. Algorithm parameters are: Match/Mismatch Scores = 1, Gap Costs = −4, similarity length is more than 50 nucleotides. A similarity search was carried out for 1000 randomly selected PPS from all the ones we found in *C. annuum* genome. We took only those similarities for which the region from −500 to −200 below *TSS_pps_* has no similarities with RNA from the transcriptome. A total of 31 PPS were found to meet all the conditions in six transcriptomes. Of these, 1 PPS are promoters that belong to annotated genes.

We then selected 1000 random positions for *TSS_pps_* in the *C. annuum* genome so that the distance between neighbors was greater than 600 bases. This was denoted as *TSS_r_*. For *TSS_r_*, we repeated the similarity search in t01-t06 transcriptomes that we did above. In this case, 17 similarities were found. Let’s put forward two hypotheses. Hypothesis *H*_0_ assumes that *p_pps_* = *p_r_*. Hypothesis *H*_1_ is that *p_pps_* ≠ *p_r_*. In our case, *p_pps_* = 31/1000 and *p_r_* = 17/1000. As a measure of the discrepancy between these two hypotheses, we used [38]:(9)$$U=\frac{m_{1}/n_{1}-m_{2}/n_{2}}{\sqrt{\frac{m_{1}+m_{2}}{n_{1}+n_{2}}(1-\frac{m_{1}+m_{2}}{n_{1}+n_{2}})(\frac{1}{n_{1}}+\frac{1}{n_{2}})}}$$

In our case, *m*_1_ = 31, *m_2_* = 17, *n*_1_ = *n*_2_ = 1000. If we substitute these values, then we get *U* = 2.07. We choose the probability that the null hypothesis is true α = 0.05. In this case, *U*_critical_ = 1.96 [38]. Since $\left|U\right|>U_{critical}$, then the hypothesis *H*_0_ is rejected and with a probability of more than 95% we can assume that *p_pps_* ≠ *p_r_*. These results indicate that about 1,4% of PPS may be active in the six transcriptomes studied and most of them are not associated with already annotated genes.

It is important to note that PSS classes were created for *Z* > 5.0 (Section 3.1), whereas PPS searches were performed for *Z* > 6.0 (Section 3.3), which explains why Table 4 contains only 41% of the 16,285 promoter sequences used to create the promoter sequence classes in Section 2.1. The difference in threshold *Z* results in the detection of more PPSs than can be found with the same number of false positives.

In total, 825,136 PPSs were identified, which is slightly less than expected considering that the number of PPSs detected in the rice genome of 4 × 10^8^ bp is 140,000 [27], which means that at the same PPS density in the *C. annuum* genome, we should have found approximately 1,050,000 PPSs.

In search for PPSs in the *C. annuum* genome, we used the MAHDS web tool at http://victoria.biengi.ac.ru/mahds/auth (accessed on 31 March 2022). MAHDS is a method for calculating multiple alignment of highly divergent nucleotide sequences, which can build multiple alignments for sequences that have accumulated up to 4.4 mutations per nucleotide [26]. It is important to note that this method allowed us to identify statistically significant similarities within promoter classes and get very few false positive results. For randomly shuffled chromosome sequences, the number of false positives is only 1068, which corresponds to approximately 3 × 10^−7^ per nucleotide and is about three orders of magnitude less than that required by other mathematical algorithms used to search for promoter sequences [17,19,31,32,33,34,35].

We studied both forward and reverse strands and found 92,617 statistically significant reverse (or mirror) PPSs with *Z > Z_0_*, which means that there were still no more than 10.1% false positives. However, it has been estimated that about 11% of promoters are bidirectional [29,39], which can lead to mirror symmetry in promoter sequences. If we apply this proportion to the 825,136 PPSs identified in our study, it would amount to about 90,000 of mirror sequences, which is close to the number of intersections detected between PPSs and mirror-symmetrical PPSs: 83,869. Thus, the 10.1% value is the maximal estimate for false positive results that could be produced by MAHDS, whereas in fact, this rate is likely to be 5–20 times lower.

It is noteworthy that, in Figure 1 and Figure 2, positions from 1 to 500 nt in multiple alignment of PPS classes are not random. However, the regions from 501 to 600 nt are more conserved, which could be due to triplet periodicity—a universal statistical feature of the gene coding regions suggesting that the usage of codons is highly nonrandom [40].

In our previous studies, we investigated the presence of PPSs in the genomes of *A. thaliana* [26] and *O. sativa* [27]. The *C. annuum* genome is the third we analyzed for PPSs. At the moment database contains PPSs from *C. annuum* and *O. sativa* genomes. In the near future we want to expand the database to include more genomes for both plants and mammals, including PPS from the human genome.

Cumulatively, our studies indicate that classification of promoter sequences makes it possible to identify about 5–30 times more PPSs compared to the number of promoter sequences near already annotated genes. The reason for such a large difference in the number of actual promoters and PPSs could lie in that there are unannotated genes, such as those encoding microRNAs, which are also transcribed by RNA polymerase II [41,42,43]. This notion is supported by the results obtained using the TSSFinder program, which found TSSs in over 50% of the PPSs identified with MAHDS. Furthermore, PPSs can be detected in dispersed repeats and transposable elements, as in case with the rice genome [27]. Since these sequences contain TSSs, it is logical that we could also identify PPSs there.

The *O. sativa* genome is about eight times smaller than the pepper genome. Moreover, 145,277 PPSs have been isolated in the *O. sativa* genome [27]. Considering the difference in genome size, about 1.16 million potential promoter sequences can be expected for the pepper genome. That is even more than what we found in the pepper genome. Of these 145,277 potential promoter sequences in the rice genome, only 37,390 PPSs were found in previously unannotated sequences. If we use the same proportion for pepper, then approximately 610,000 of the 825,136 detected PPSs will be from various dispersed repeats, transposons, and viruses. And only 212,365 may be in regions that do not contain known genes, transposons, and various dispersed repeats and viruses. It can be assumed that these PPS may be associated with the formation of de novo promoters and genes, as noted earlier [44,45,46].

In conclusion, our results indicate the effectiveness of the MAHDS method for the comprehensive, genome-wide detection of statistically significant PPSs with a very low rate of false positive results. The created databank could be useful for genetic engineering and biotechnology.

## Figures and Tables

**Figure 1 biology-11-01117-f001:**
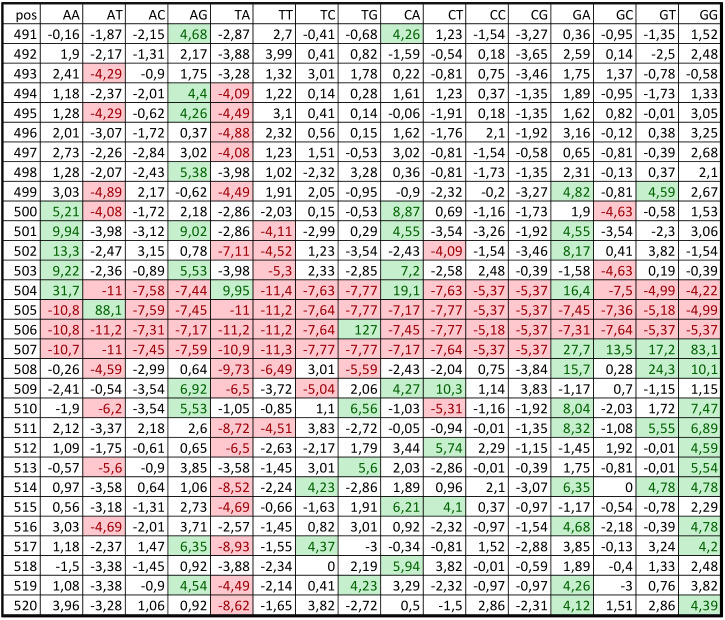
Part of the *PWM’* for the first *C. annuum* promoter class. Elements with values <−4 and >4 are highlighted red and green, respectively.

**Figure 2 biology-11-01117-f002:**
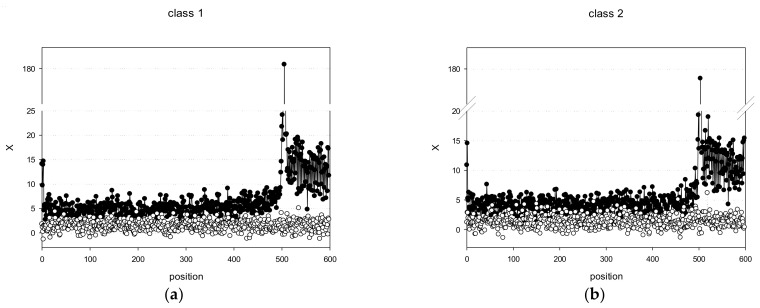
Profile diagrams of *C. annuum* promoters of class 1 (**a**) and class 2 (**b**). Black and white circles indicate *X(j)* for promoter sequences and random sequences, respectively.

**Figure 3 biology-11-01117-f003:**
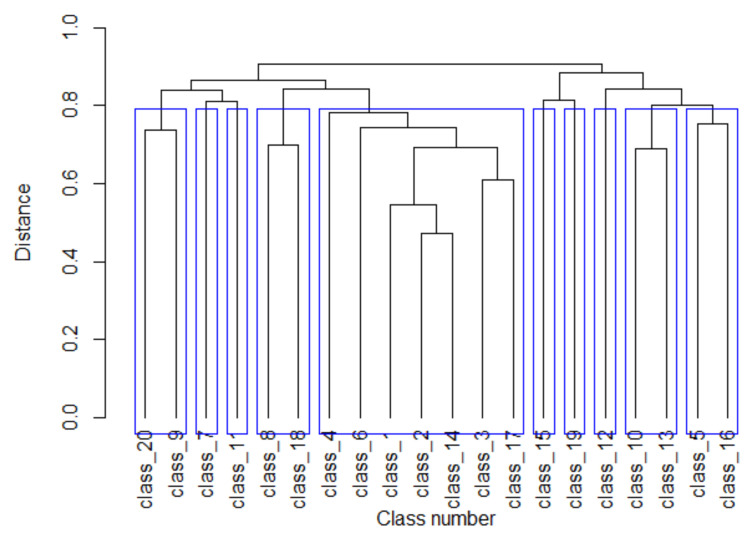
Cluster dendrogram of *C. annuum* promoter classes. Blue rectangles indicate clusters of promoter classes obtained at the association level of 0.8.

**Figure 4 biology-11-01117-f004:**
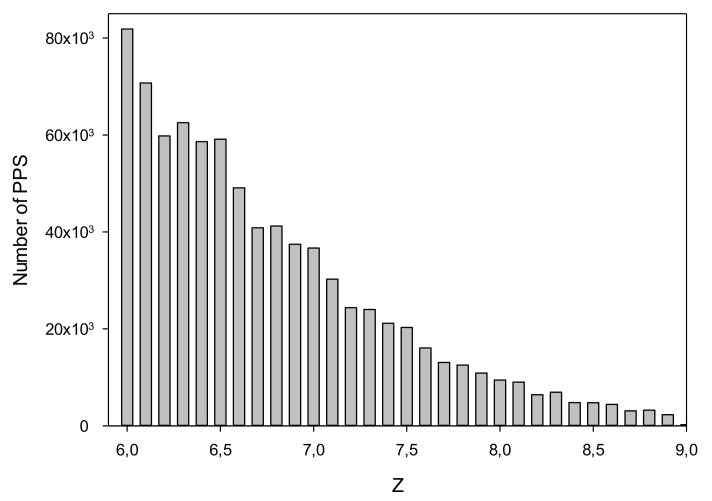
Histogram Z for PPS.

**Table 1 biology-11-01117-t001:** Numbers of elements in the created classes of *C. annuum* promoters.

Class №	Number of Elements	Class №	Number of Elements
1	5402	11	203
2	1976	12	176
3	993	13	171
4	695	14	159
5	515	15	141
6	400	16	129
7	390	17	118
8	321	18	118
9	230	19	117
10	214	20	106

**Table 2 biology-11-01117-t002:** Number of PPSs in the *C. annuum* genome for different *Z_0_* levels.

Results	Z Level				
	≥ 5.0	≥ 5.5	≥ 6.0	≥ 6.5	≥ 7.0
Real sequences	1,679,534	1,242,664	825,136	491,647	263,864
Random sequences	20,490	5064	1068	221	44
FDR	1.21%	0.41%	0.13%	0.04%	0.02%

**Table 3 biology-11-01117-t003:** Matches of the detected PPSs with promoter regions of the annotated genes in the C. annuum genome (Z ≥ 6.0).

Chromosome №	Annotated Genes (*n*)	Matches in the Strands of Annotated Promoters and PPSs (n) *				Total Matches (*n*)	% of Matches
		++	−−	+−	−+		
1	2161	431	387	53	38	909	42%
2	1759	294	420	56	25	795	45%
3	1988	426	325	21	50	822	41%
4	1269	220	286	31	29	566	45%
5	1054	240	145	11	32	428	41%
6	1555	425	185	14	110	734	47%
7	1231	223	223	22	28	496	40%
8	649	113	124	9	15	261	40%
9	1104	187	241	58	14	500	45%
10	1075	223	180	19	49	471	44%
11	1166	187	198	22	30	437	37%
12	1274	215	222	26	31	494	39%
Total	16,285	3184	2936	342	451	6913	42%

* + and − indicate forward and reverse strands, respectively; the first and second characters refer to the annotated promoters and PPSs, respectively.

**Table 4 biology-11-01117-t004:** Numbers of TSSs identified in the PPSs of the *C. annuum* genome using TSSFinder with different training sets.

Chromosome №	*A. thaliana*		*O. sativa*		*H. sapiens*	
	All PSSs	TATA-Containing PSSs	All PSSs	TATA-Containing PSSs	All PSSs	TATA-Containing PSSs
1	46	9	8	0	16	0
2	38	3	7	0	15	0
3	60	13	17	0	12	0
4	37	8	6	0	15	0
5	61	13	13	0	21	0
6	72	18	11	0	16	0
7	49	10	13	0	19	0
8	60	10	10	0	18	0
9	41	4	7	0	21	0
10	65	9	10	0	25	0
11	52	7	13	0	15	0
12	48	11	10	0	20	0
Total	629	115	125	0	213	0

## Data Availability

The data presented in this study are available in this article and in the Appendix A.

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
