# Peer review of "Database of Potential Promoter Sequences in the Capsicum annuum Genome"

_biology, 2022, doi:10.3390/biology11081117_

Round 1

Reviewer 1 Report

In this paper, authors searched for potential promoter sequences in pepper genome. They used a new mathematical method for multiple alignment of highly divergent sequences.  Yet, this so called new method has been used at least in their two papers.  It would be better, if authors could clearly describe what's new in this paper. 

Author Response

We would like to thank the Reviewer for the constructive input and careful assessment of our manuscript.

Reviewer 2 Report

In this paper, a database was created (dbPPS) of Capsicum annuum genome promoter sequences with
new method for multiple alignment. The database can be viewed at the URL link 
http://victoria.biengi.ac.ru/cgi-bin/dbPPS/index.cgi .
I was tested the database with a several queries, it works as expected.
I noticed a few minor issues in this preliminary PDF document, almost all URL links are broken (it is not possible to click on them to open a web page) and on page 7 there is a typo in formula (5), the Chinese character is shown.

Perhaps it would be good idea to attach the source code of the programs used for this 
analysis (if exist), as this would gain this paper in popularity. It would be great if other scientists could 
do the same analyzes but on different genomes.
For example, if you did the analysis, mathematical calculations and database develop with some programming language, 
I think it's interesting for other scientists to see that source code in some public code repository (e.g. github),
and possibly continue to improve/contribute the source code there.

Author Response

(The authors gave the same response as above.)

Reviewer 3 Report

The main claim of the manuscript (as reflected in the title and the abstract) is that the authors have prepared a database with promoter sequences predicted in the pepper genome. However, the description of the database is rather limited and, unfortunately, this reviewer has not been able to access the link to the database (supplied by the authors in the text). The paper is mostly focused on the application of the author's mathematical method to a new dataset (the genome sequence of pepper), but the method itself is not new, as it has already been applied to a couple of other genomes, as the authors themselves cite in the text. A list of predicted promoter sequences is supplied as supplemental material, but it is out of context, which makes it difficult to assess if such sequences are in the vicinity of known genes or other elements (e.g. transposons etc). 

I found the mathematical aspects somewhat difficult to follow, although I am assuming they are correct, as their method has previously been subjected to peer review.  Equation (5) does not read well in the PDF file. The use of X for X²(j) and X(j), and their relationship, on page 6, is also somewhat confusing.

I'm somewhat skeptical on the authors' claim that they have identified 825,136 promoters, with a very low rate of false positives. Beyond their predictions, the authors do not supply any experimental evidence that these promoters are real (e.g. by testing the expression of adjacent sequences, or by showing their evolutionary conservation in related species). I would suggest that the authors try to confirm the activity of their predicted promoters with  data from RNA-seq or similar available experiments. One additional concern is if the random sequences provide a good baseline to assess the amount of false positives, particularly as the matrices were not prepared from random sequences.

Author Response

The authors are very grateful to the referee for the efforts in reviewing our work. We tried to take into account all the remarks of the reviewer, which, undoubtedly, improved our manuscript.

Round 2

Reviewer 3 Report

I have taken the time to carefully read the revised version of the manuscript and the authors' response to the previous evaluations.  While I realize that the authors have made an effort to address my concerns, I am still somewhat skeptical about the authors claims of a low number of false positives. The authors have now performed an experiment using  RNA-seq data, which is presented in lines 491-522 of the manuscript. These new results should be presented in the manuscript as Results (possibly in the Materials and Methods section too), and not just generically described in the Discussion section. To me, the main problem is the huge gap between the number of known genes and predicted promoter sequences. In the authors' words, the RNA-seq data indicate that 1.4% of the PPS "may be active". If we consider together these "new" genes, and the number of annotated genes (~30,000), we are still left with more than 700,000 orphan PPSs. Of course, some of them might match the sequence of transposable elements, but does that mean that they correspond to promoters, even if those promoters are considered to be "potential"? To me, in the absence of additional evidence, the most plausible explanation is that they are simply false positives, despite the authors' intense efforts to demonstrate the opposite. As I am not a mathematicians, it is hard for me to explain why the experiments shown lead to such low estimates of the number of false positives --perhaps the frequency distribution for the dinucleotides in the original dataset is different from that in the shuffled or "complementary" sequences, and this might lead to differences is the amount of detected sequences. Have the authors checked on this?

I have now been able to access the database. In my opinion, it would be more useful if the information was presented visually (using a genome browser), presenting the location of the predicted PPSs and the known genes and transposable elements in different tracks.

If the manuscript is eventually accepted, I would ask that some claims in the text are toned down. As an example, please change "identification of all promoter sequences" to "prediction of putative promoter sequences" (on line 16, in the abstract. 

Similarly, on line 25, the authors make the hard-to-demonstrate claim that their tool "allows genome-wide identification of all promoter sequences, including those of unannotated genes and transposons". If this was the case, I feel they should take advantage of this capability of their tools and report the annotate those new genes and transposons. 

Round 3

Reviewer 3 Report

I realize the authors and I hold widely divergent views and, at this point, it should be the editor who adopts a decision based on the reports of the other reviewers and my own one. I will only comment on two of the authors' points:

In point #1 of their reply, the authors claim that the text presented in lines 491-522 is " just a comparison of our results with the experimental data obtained by other authors". Well, even if the raw data were obtained by different authors, the authors used bioinformatics software (e.g. Megablast) to identify hits between the Illumina reads and arbitrarily selected 100-nt long sequences located at positions adjacent to their predicted transcription start sites. To me, Megablast is the wrong alignment tool for this. A better approach would involve using a spliced read mapper, such as STAR or HISAT2, to map the reads to the genome sequence. Indeed, the conclusion of this bioinformatic experiment (yes in my view this is an experiment) is given in the Discussion:  "about 1,4% of PPS may be active in the 6 transcriptomes studied and most of them are not associated with already annotated genes", which is in line with my skepticism.

In a somewhat confusing sentence of their reply, the authors incorrectly state that there are "hundreds of thousands" of microRNAs "per genome size". I think this is incorrect: the number of validated microRNAs in plant genomes is three orders of magnitude lower, in the order of hundreds. However, it might be true that some computer programs, which predict microRNAs based on conserved characteristics, can possibly identify hundreds of thousands of sequences that can fold forming a stem-loop structure. Of course, the creators of such software could always refer to such predictions as "potential" microRNAs, but such denomination would not make the work more relevant and those sequences would still not be real microRNAs. 
